# CNN-Based Defect Inspection for Injection Molding Using Edge Computing and Industrial IoT Systems

**Hyeonjong Ha**  and **Jongpil Jeong** * 

Department of Smart Factory Convergence, Sungkyunkwan University, Suwon 16419, Gyeonggi-do, Korea; wnajrqkq94@naver.com
* Correspondence: jpjeong@skku.edu

**Abstract:** Currently, the development of automated quality inspection is drawing attention as a major component of the smart factory. However, injection molding processes have not received much attention in this area of research because of product diversity, difficulty in obtaining uniform quality product images, and short cycle times. In this study, we proposed a defect inspection system for injection molding in edge intelligence. Using data augmentation, we solved the data shortage and imbalance problem of small and medium-sized enterprises (SMEs), introduced the actual smart factory method of the injection process, and measured the performance of the developed artificial intelligence model. The accuracy of the proposed model was more than 90%, proving that the system can be applied in the field.

**Keywords:** defect detection; edge computing; smart factory; CNN; injection molding



## 1. Introduction

Injection molding has been widely used in the manufacturing industry, from small companies to major companies. The production of the injection begins with mold design and continues with raw material injection, injection molding, product emissions, visual inspection and quantification, packaging, and delivery. Defect inspection is very important in these injection molding processes, because it can reduce the risk and cost of providing defective products to customers. The manufacturer does a final defect inspection before delivery to the consumer. Many small and medium-sized enterprises often do quality checks manually. Such manual inspection is prone to human errors. In addition, continuous training of field professionals in reproducibility verification to bring each person to the same level is essential. Repeating this process is so costly that the risk of financial losses throughout the industry has increased the urgency of automating surface defect detection and expanding it to manufacturing. Worker fatigue is caused by repetitive work. To address these issues, many studies have been conducted on automation and defect detection [1–4].

Among these studies, research on building a smart factory using the Internet of Things (IoT) is actively under way. Its purpose is to make smart factories, factory equipment, and sensors (IoT) collect and analyze data in real time, and see (observability) all situations of factories at a glance. A smart factory refers to a factory that can control itself. IoT-based machines extend the boundaries of smart factories to demonstrate new possibilities for manufacturing.

Edge computing is defined as long-term cloud computing (CC) where data are calculated near the edge of the network where data are generated. Applying the latest computational approach, DL (Deep Learning) has been widely used for intelligence in various fields such as image classification [5], semantic segmentation [6], and image compression [7]. DL's self-learning and compression capabilities allow it to automatically learn the characteristics of the input data hierarchically, emphasizing hidden and anomalous patterns. As a result, DL can be the most widely used quality inspection technology at present. The development of deep learning technology has made great achievements, especially in

the field of image detection. In particular, convolutional neural networks (CNN) model achieved higher accuracy than humans perceived in the ImageNet Challenge [8].

However, automatic visual inspection has several problems when applied to the injection process. First, problems occur when inspecting large quantities of products at high speed. Because several products are released at once in mass production, we need an inspection model that can be processed simultaneously and quickly. Second, data imbalance is a problem. 99% of the data of products produced in the injection molding process are normal data, and only 1% are abnormal data. Abnormal data are insufficient compared to normal data, so we need to find a way to solve it. Finally, defect detection is carried out in a plane. Since the defect inspection is done on only one side, the quality inspection must be done on the parts that are not photographed.

Therefore, in this paper, we present a novel method called a defect inspection framework based on deep neural networks for injection molding in IoT Systems with Edge Computing. In the training process, data augmentation techniques are initially used to improve the stability and performance of deep learning. Various data augmentation methods have been studied and applied to solve the problem of lack of data in the field. A typical example is the medical field. Data augmentation is used to construct big data in the medical imaging field [9,10], where it is difficult to obtain enough data with personal information, such as synthesis using generative adversarial networks (GANs) [11], as well as methods such as rotation and flip. Various studies are underway, including ones on finding a method to increase data. Data augmentation does not significantly undermine the information contained in the original data, and improves learning performance with only a little of the original data by increasing data with the same contextual characteristics. When the object produced by the sub-motor rotates, it is shot through the vision camera and to the edge box, which presents a quality check automation model that detects faults in the Edge Box and transfers the index of product-fault data to the programmable logic controller (PLC).

The paper is organized as follows. Section 2 describes related work about CNN and edge computing. Section 3 details the overall defect detection system and model for molding injection industry. Section 4 describes the evaluation indicators and results from the experiment. Finally, Section 5 presents the conclusion.

## 2. Background and Related Work

### 2.1. Defect Detection for the Injection Molding Process

Various studies have been implemented to understand shrinkage and to control the dimensions of injection molding. Kramschuster et al. [12,13] applied an experimental design to conduct quantitative studies of the shrinkage and warping of fine-porosity and existing injection molds. Kwon et al. [14] studied anisotropic contraction in injection molding of amorphous polymers considering the pressure-volume-temperature equation of state, molecular orientation, and elastic recovery. Kurt et al. [15] investigated the effect of packing pressure, melting temperature, and cooling time on shrinkage of injection molds. Santis et al. [16] explored the effects of suppression, time, and geometric constraints on the contraction of semi-crystalline polymers with strain gauges. Chen SC et al. [17,18] applied gas backpressure to reduce the shrinkage of parts during injection molding. Qi et al. [19] found that mixing of polypropylene copolymers can effectively reduce the molding shrinkage of isletic polypropylene. Lucyshyn et al. [20] identified the transition temperature used in injection molding simulations (i.e., moldflow) to calculate contractions. Wang et al. [21] used artificial neural network (ANN) simulations to evaluate the effectiveness of molding parameters on molding shrinkage. Abdul et al. [22] developed a shrinkage prediction of injection molded dense polyethylene parts using the Taguchi approach and ANN. Sidet et al. [23] and Guoet et al. [24] studied the tensile strength and shrinkage of thermoplastic complexes in injection molding. Kc et al. [25] applied the Taguchi approach to reduce shrinkage of injection-molded hybrid biocomposites. Mohan et al. [26] conducted a comprehensive review of the effects of molding parameters on the strength, shrinkage, and bending of plastic parts. All of these studies are very useful for improving the under-

standing of molding shrinkage and optimizing machine parameter settings. Furthermore, Mirjavadi et al. [27–30] investigated the vibration and thermal behavior of functional class materials considering vibration and material distribution in the study.

*2.2. CNN*

Deep-learning techniques, which learn by building deep neural network layers, have evolved rapidly because of the massive number of data and amount of computation associated with GPU performance development. Let us look at deep neural networks to understand the behavior of deep learning. The input layer that accepts input data in this network predicts the value of the end result. The output layer extracts features consisting of hidden layers with layer stacks of different depths between the input layer and the output layer. The data learning process is a cost function that feeds input data to the input layer and the hidden layer, showing the difference between the output values predicted by the final output layer and the target label of the input data. Reverse propagation is done on differences in the cost function (Gradient), and the weights of all layers are gradually updated.

CNN is a type of artificial neural network that can be easily applied to video and image. When input images are given to the input layer, convolutions are executed sequentially for overlapping parts by small filters. One filter has weights of that size and does weight learning to extract features of the image. The filter moves horizontally and vertically in the input image, doing convolution and activation function operations, extracting features, and displaying Feature Map Yield. This computational method is similar to image convolutional computation in the field of computer vision. Deep neural networks of these structures are called CNNs.

Since Lecun et al. [31] developed CNNs, several defect detection models have been developed for industrial products. CNN models have made breakthroughs in computer vision and are widely used for various applications such as image classification [32], image segmentation [33], and object tracking [34]. Surface-defect detection [34–36] identifies cosmetic defects in fabrics, metals, woods, and plastic products by using image-processing technology. Targets may differ, but surface-defect detection is a feature extraction process used to identify anomalies that can be distinguished from textures. Algorithms that extract features from textures to detect surface defects can be defined according to four categories [37]: statistical, structural, filter-based, and model-based approaches. Statistical and filter-based approaches have been widely used. For example, histogram properties classified by statistical approaches have been applied to various studies [38,39] and have worked well at low cost and effort. Among the co-space/space frequency methods classified as a filter-based approach, the Gabor transformation [40] (using modified Gaussian filters) is widely used, because it is similar to the human visual system. After CNN was developed, filter kernel-based neural networks were proposed, and CNN-based feature extraction techniques were quickly developed in the fields of image processing and machine learning research. Ren et al. [41] applied a general deep learning approach based on CNN models for automatic surface examination. Star et al. [42] used the modified CNN model triplet network to teach Deep Matrix to do anomaly detection for industrial surface examination. Wang et al. [43] proposed a CNN-inspired dual joint detection model to classify industrial surface inspections. Tao et al. [44] proposed a cascaded autoencoder architecture based on CNNs to segment and localize multiple defects in industrial product data. Furthermore, many researchers have proposed various robust CNN-based models [45,46] to address image classification problems or defect location problems for various industrial surface defects. Recently, concrete crack detection research using CNN-based models has been actively conducted. Deng et al. [47] applied a temporary fast region-based CNN (Faster RCNN) to distinguish between handwritten scripts and cracks in concrete surfaces. Chun et al. [48] detected cracks in concrete surfaces using a light gradient boosting machine (LightGBM) considering pixel values and geometric shapes. You Only Look Once (YOLO),

VGG Net, Inception Net, and Mask R-CNN have been frequently applied to detect concrete cracks in civil and infrastructure engineering studies [49].

### 2.3. Edge Computing

Because data are increasingly generated at the edge of the network, it is more efficient to process data there. Previous work has been introduced to the community, such as micro data centers [50,51], cloudlet [52], and fog computing [53]. This is why cloud computing is not always efficient in processing data when it is generated at the network edge. This section lists some of the reasons why edge computing is more efficient than cloud computing in some computing services and then provides definitions and an understanding of edge computing. Edge computing can do computations at the edges of a network on downstream data that replace cloud services and upstream data that replace IoT services. Here we define "edge" as all computing and network resources along the path between the data source and the cloud data center. For example, a smartphone is the edge between a body object and a cloud, a gateway to a smart home is the edge between a home object and a cloud, and a micro data center and a cloudlet are the edge between a mobile device and the cloud. The rationale for edge computing is that computing should occur near a data source. From our perspective, edge computing is interchangeable with fog computing, but edge computing is more focused on the object side, whereas fog computing is more focused on the infrastructure side. Edge computing can have as much an effect on our society as does cloud computing.

### 2.4. Industrial IoT Systems

IoT, an emerging technology sector, has drawn keen attention from governments, research institutes, and businesses. The term IoT was coined in 1999 by Kevin Ashton, who aimed to connect different objects over a network. Currently, "things" can be RFID (Radio Frequency Identification) tags, sensors, actuators, mobile phones, lightweight wearables, and even uniquely identifiable virtual entities [54]. Although the definition of "things" has changed as technology advances, the essential attributes of interacting with each other and working with neighbors to achieve common goals remain intact without human intervention. The expected interaction between the huge number of interconnected objects, objects and high-performance computing, storage centers, and increasingly intelligent IoT devices opens up new opportunities for creating smarter environments [55]. Industrial IoT uses IoT technology to collect real-time data, control manufacturing environments, and monitor environmental metrics such as hazardous gases, temperatures, and humidity and fire alarms and can significantly improve manufacturing efficiency and reduce enterprise costs. Therefore, interest in using IoT technology in various industries is increasing. Numerous industrial IoT projects have been undertaken in areas such as agriculture, manufacturing and processing industries, environmental monitoring, and mining safety monitoring. Industrial IoT devices are sensors, controllers, and special equipment that range from small environmental sensors to complex industrial robots and can accommodate primarily harsh and complex industries [56,57]. IoT applications focus on collecting and processing sensing and decision data in industrial environments and providing many notifications [58]. IoT used in a Smart Factory (or Industry 4.0) by integrating new technologies in production processes could improve working conditions (an example could be the support of a robot to the human operator) as well as safety and productivity in an industry [59–62].

## 3. CNN-Based Defect Inspection for Injection Molding

### 3.1. System Architecture

The proposed model of overall architecture is composed as shown in Figure 1. Image data are acquired by means of a vision camera that scans the photographing unit and sends it to the edge box. The defect inspection is done in the edge box. If a defect is detected, the number of the defective cell is transmitted to the PLC, which plays the role of removing defective products from the PLC.

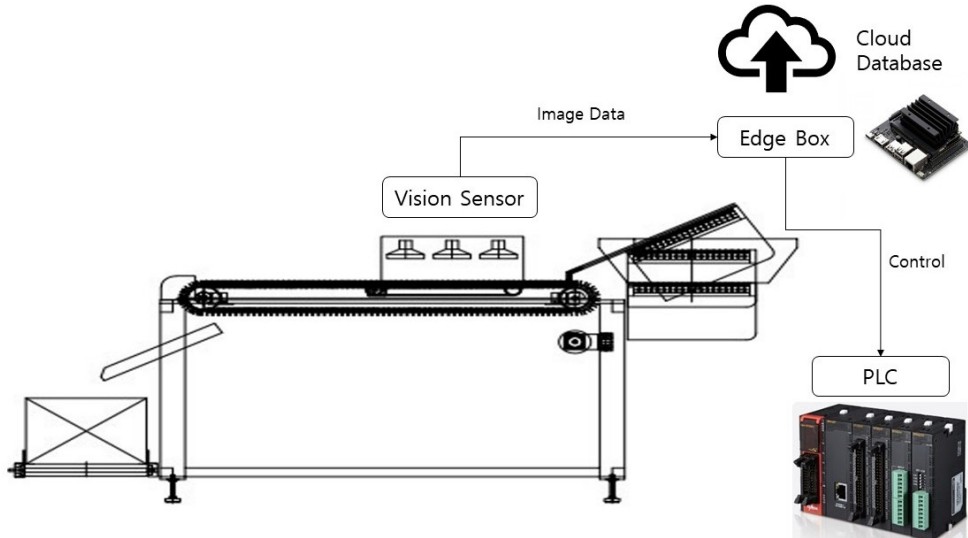

**Figure 1.** System Architecture.

Image data are acquired using lighting and a GigE vision camera. Lighting minimizes how much the difference between day and night affects quality inspection. It is configured in the form of a conveyor belt that connects the rails in a cylinder. The advantage of these rails is that the product to be inspected is rotated while the product is being inspected so that one can inspect the quality of all surfaces of the object rather than one. In system design, the product is inspected twice, improving the existing deep learning inspection method by means of CNN. Figure 2 is a picture of the product taken by the vision sensor on the rails.

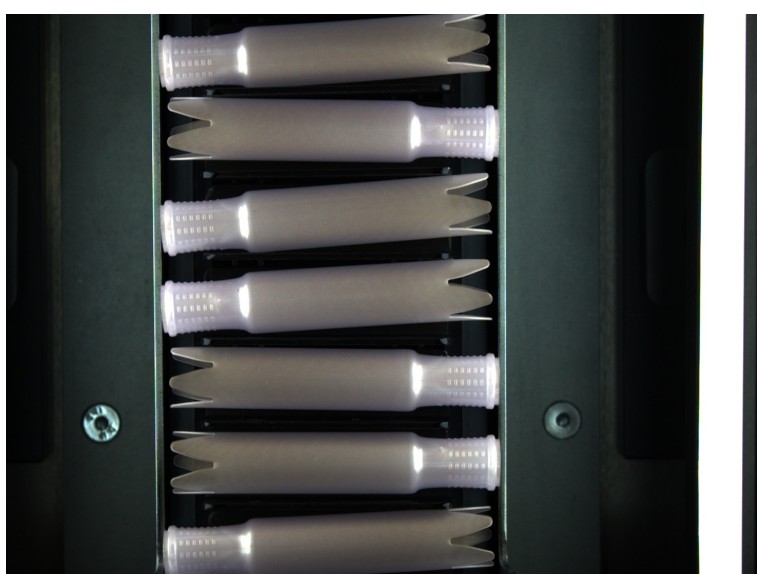

**Figure 2.** Products on the Rail photographed with a Vision Sensor.

The algorithms done in the edge box are summarized in Table 1. First, when a raw image comes by means of the vision sensor, it is cropped as an image of a product for defect detection. Then, it does defect detection and finds out how many times the product in the cell was defective. These data would be transferred to the database via the cloud. Finally, it communicates with the PLC. If the time from the $n$-th cell to the discharge port is calculated and transmitted to the PLC, the defective product is discharged in the final quality inspection. We designed the automated system.

**Table 1.** Algorithms performed in Edge Box.

|  | Input | Output |
|---|---|---|
| Algorithm 1 | Raw image | Cropped image |
| Algorithm 2 | Cropped image | The number of cell that is defect |
| Algorithm 3 | The number of cell that is defect | Time from $n$-th cell to discharge |

### 3.2. Defect Detection

Deep learning builds up many concealed layers to increase the parameters to increase the model's expressiveness. Training many parameters properly requires a huge number of training data. However, extracting enough data from the actual working conditions is not easy. In addition, data should be diverse enough to maintain high quality and reflect reality. Using deep-learning models that do not have sufficient training data to train parameters usually results in underfitting problems. Therefore, data augmentation [47,48] allows us to increase the absolute number of data even in small data set regions, thereby applying artificial changes to the data to obtain new data. Data augmentation can handle unexplored inputs and improve the generalization of deep-learning models. An important point about data augmentation is meeting domain knowledge to maintain existing labels when creating new data. It also does not change the data label because of minor changes. Data augmentation is often used in images, but data augmentation is applied to time-series data.

In this paper, we have used three data augmentation techniques, all of which were based on the fact that a slight change in the action point can keep the label. First, a Resize and Rescaling technique changes the size of the image. Second, we propose a system that can inspect all product sides, not flat-image product inspection. Some frameworks do not provide a function for vertical flips, but a vertical flip is equivalent to rotating an image by 180 degrees and then doing a horizontal flip. Finally, image dimensions may not be preserved after rotation. Rotating the image by finer angles will also change the final image size.

For the detection of defects on molding products, we propose a novel CNN architecture. The data extracted by means of the vision sensor arrives as input to the inspection model in two dimensions. Image data are processed in grayscale. The architecture of the proposed CNN architecture for defect detection is shown in Figure 3. Data that were rescaled were of size $300 \times 300$. The input data were fed into a layer with three differently sized convolution kernels. The first convolutional layer had a $7 \times 7$ convolutional kernel. The second and third convolutional layers each had a $3 \times 3$ convolutional kernel. The maxpooling layer is behind each convolutional layer and is $2 \times 2$. After passing through the three convolutional layers and the maxpooling layer, data enter the flattened layer. They are then compressed by means of the Dense Layer. To avoid overfitting, we applied the dropout technique and set the dropout rate to 0.2. After that, the architecture would be completed with a softmax layer at the end for defect detection. Table 2 summarizes the architecture of the CNN model used in the paper.

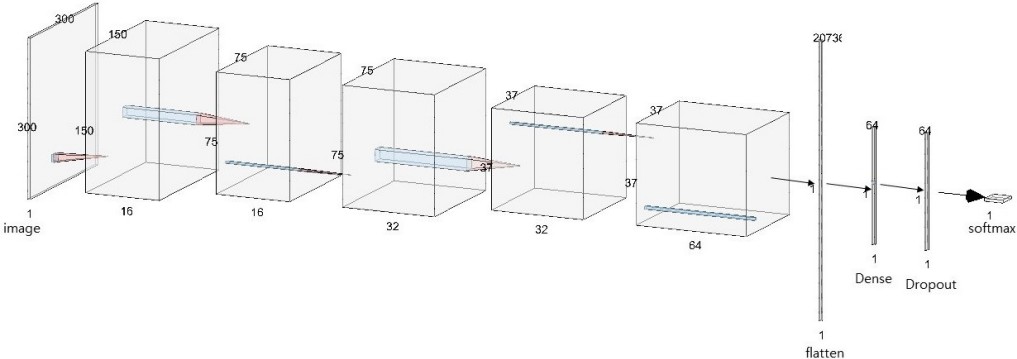

**Figure 3.** Proposed Architecture for Defect Detection.

**Table 2.** Summary of Proposed Architecture.

| Layer Name | Output Size | Network | Connected to |
|---|---|---|---|
| Input Layer | (300 × 300) | Conv2D | |
| Conv Layer1 | (150 × 150 × 16) | Conv2D, kernel size = 7 × 7 | Input Layer |
| Pool Layer1 | (75 × 75 × 16) | Maxpooling2D, size = 2 × 2 | Conv Layer1 |
| Conv Layer2 | (75 × 75 × 32) | Conv2D, kernel size = 3 × 3 | Pool Layer1 |
| Pool Layer2 | (37 × 37 × 32) | Maxpooling2D, size = 2 × 2 | Conv Layer2 |
| Conv Layer3 | (37 × 37 × 64) | Conv2D, kernel size = 3 × 3 | Pool Layer2 |
| Pool Layer3 | (18 × 18 × 64) | Maxpooling2D, size = 2 × 2 | Conv Layer3 |
| Flatten Layer | (20,376) | Flatten | Pool Layer3 |
| Dense Layer | (64) | Dense | Flatten Layer |
| Dropout Layer | (64) | Dropout, rate = 0.2 | Dense Layer |
| Softmax | (1) | Dense | Dropout Layer |

## 4. Experiment and Result Analysis

In this section, we describe the selection of an indicator to evaluate the proposed system and to conduct the experiment and then discuss the results.

### 4.1. Experiment Environment

The hardware used in this study consisted of a computer with an Intel Core i7-8700 K processor, GTX 1080 Ti, and 12 GB RAM. Therefore, it was possible to reduce the training time and improve the performance, unlike the capabilities of previous equipment. The result of the algorithm may vary depending on the environment of the experiment. The system specifications used for the experiments are listed in Table 3.

**Table 3.** System Specifications.

| Hardware Environment | Software Environment |
|---|---|
| CPU: Intel Core i7-8700 K, 3.7 GHz, Six-core twelve threads, 16 GB GPU: Geforce GTX 1080 Ti | Windows TensorFlow 2.0 framework Python 3.7 |

During the experiment, we collaborated with a company called Telstar-Hommel. Furthermore, we used software tools called LINK5. Telstar-Hommel has 30 years experience of building assembly lines, measurement machines, and quality control systems for the automotive industry. LINK5 is Telstar-Hommel's independent Smart Factory platform created based on years of experience in building automation lines in various industries and the know-how of IT professionals. It is a solution for quality improvement and productivity enhancement by monitoring the situation occurring in the production line and managing/analyzing all generated information to improve the productivity and quality of the customer production line. It collects information that occurs throughout the plant's facilities and production in real time and provides each function in a modular fashion. As a specialized company in automation equipment for 30 years, it is possible to build a more accurate and efficient production and quality management system with knowledge of equipment and IT convergence. Using this software tool, PLC and edge box were connected, and vision sensor and edge box were connected. Furthermore, the algorithm performed in Edgebox is implemented in Python.

The vision sensor used in the above experiment is shown in the Figure 4. These vision sensors were used to collect data. Two vision sensors collected image data.

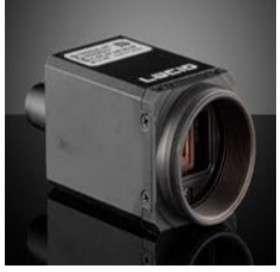 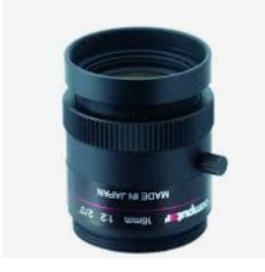 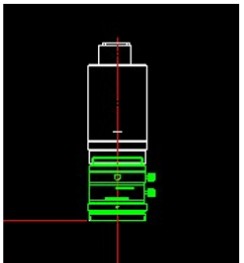

Model: TRI050S       Model: MI620-MPW2       Vision Model

**Figure 4.** Vision Sensors used in Proposed System.

Figure 5 shows the rail to be inspected by the vision sensor. Products enter the rail one by one, and the rail rotates through a sub-motor. Since this structure is photographed while the tampon applicator is rotated on the rail by a sub-motor, the system can inspect all sides of the product.

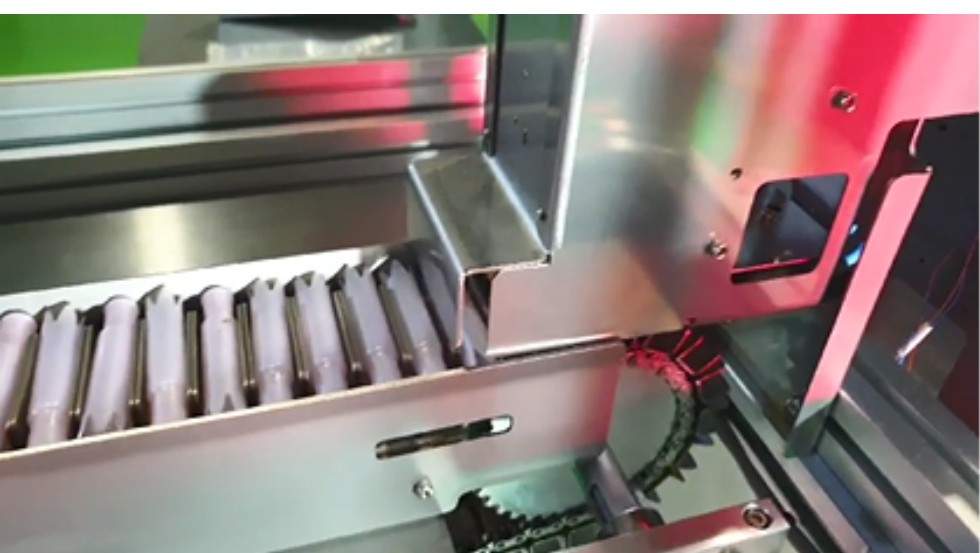

**Figure 5.** Product on rail.

### 4.2. Evaluation Metrics

We calculated the receiver operating characteristic (ROC) curve, Matthews correlation coefficient (MCC), accuracy, and F1-score to evaluate the performance of the classifier for bearing defects in noisy situations. The MCC is used in machine learning as a measure of the quality of binary and multiclass classifications. It takes into account true and false positives and negatives and it is generally regarded as a balanced measure, which can be used even if the classes are of very different sizes. The MCC equation is:

$$\text{MCC} = \frac{|\text{TP}| * |\text{TN}| - |\text{FP}| * |\text{FN}|}{\sqrt{(|\text{TP}| + |\text{FP}|)(|\text{TP}| + |\text{FN}|)(|\text{TN}| + |\text{FP}|)(|\text{TN}| + |\text{FN}|)}} \tag{1}$$

The ROC curve is a widely used method of evaluating the effectiveness of a diagnostic method. It represents the relationship between sensitivity and specificity on a two-dimensional plane. The larger the area under the ROC curve, the better the model. Sensitivity and specificity can be expressed by the following equation.

- Confusion Matrix: A matrix that shows the predicted class result compared to the actual class at once;
- Positive (=Normal Status): Normal situation that the quality manager wants to maintain (OK);

- Negative (=Anomaly): Unusual situation in which the quality manager needs to be involved (NG);
- False Positive (=Type I Error = Missing Error): A situation where AI misses when a failure occurs (FPR);
- False Negative (=Type II Error = False Alarm): A situation where AI reports a failure even though it is not a failure (FNR).

Specificity is the rate at which the model recognizes false as false. The equation is as follows:

$$\text{Specificity} = \frac{\text{TN}}{\text{TN} + \text{FP}} \tag{2}$$

Recall is the proportion of the true class to what the model predicts as true. The parameters recall and precision have a trade-off. Recall, also called sensitivity, can be expressed as follows:

$$\text{Recall (Sensitivity)} = \frac{\text{TP}}{\text{TP} + \text{FN}} \tag{3}$$

Precision is the ratio of the true class to what the model classifies as true. The equation is as follows:

$$\text{Precision} = \frac{\text{TP}}{\text{TP} + \text{FP}} \tag{4}$$

Accuracy is the most intuitive indicator. However, the problem is that unbalanced data labels can skew the performance. The equation for this parameter is the following:

$$\text{Accuracy} = \frac{\text{TP} + \text{TN}}{\text{TP} + \text{FP} + \text{FN} + \text{TN}} \tag{5}$$

The F1-score is called the harmonic mean, and if data labels are unbalanced, it can accurately assess the performance of the model. The equation is given as follows:

$$\text{F1-score} = 2\frac{\text{Precision} * \text{Recall}}{\text{Precision} + \text{Recall}} \tag{6}$$

### 4.3. Experiment and Results

In this paper, in order to verify the system, we obtained data from a small and a medium-sized business plant in the Republic of Korea. The company is producing female products, that is, tampons. The tampon applicator is one of the products produced by injection molding. In the current inspection process, the worker manually inspects the product. We used 20% of the training data as validation data during training. We collected data for the introduction of the smart factory, as summarized in Table 4. Before the training process of the model, we used the data augmentation technique. Sample images of the product are shown as Figure 6.

**Table 4.** Dataset of Proposed Model.

|  | Normal | Defect |
|---|---|---|
| Training Data | 1714 | 200 |
| Validation Data | 316 | 100 |
| Test Data | 198 | 55 |

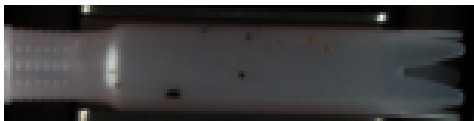 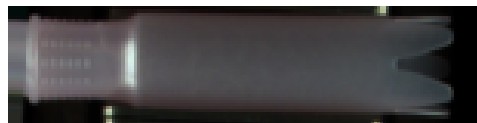

**Figure 6.** Sample Images of Product: Defect (**left**) and OK (**right**).

Figure 7 depicts the model's training process. Epoch was set to 50, and training was stopped if validation loss did not improve after 10 or more epochs were repeated. We obtained 27 epoch results. By using checkpoints, we used the model with the lowest validation loss. Figure 7 is graph showing the values of training, training accuracy, and validation accuracy as training progress.

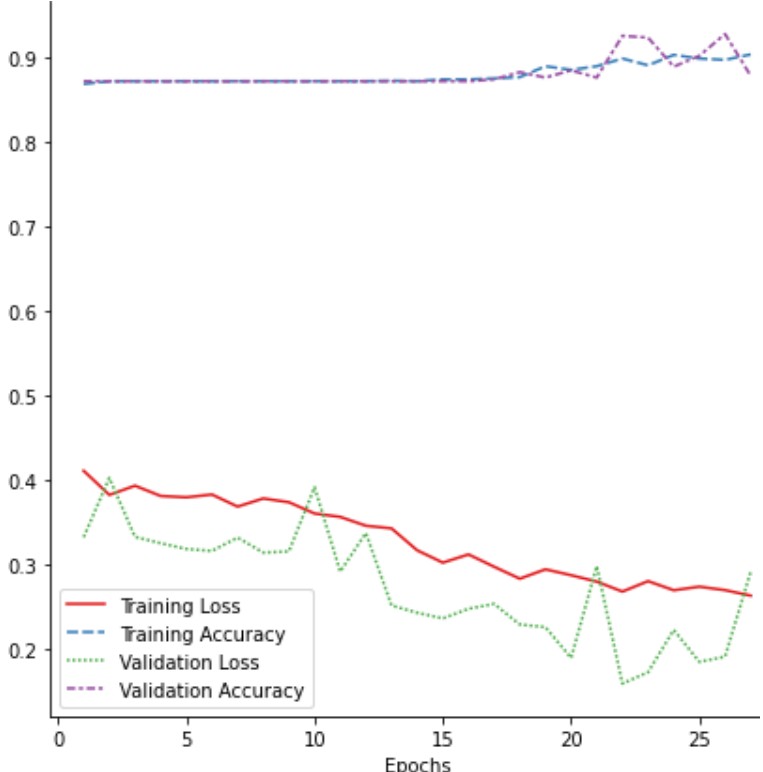

**Figure 7.** Learning History of Accuracy and Loss in Training Process.

Table 5 summarizes the proposed model's Precision, Recall, and F1-score values. It showed more than 90% accuracy, which is the development goal. The ability to predict the normal product from the normal person data is good, but the ability to accurately predict the defect data from the defect data is insufficient. The MCC score is 0.7311.

**Table 5.** Results of Proposed Model.

|  | Precision | Recall | F1-Score |
|---|---|---|---|
| Normal | 0.9581 | 0.9242 | 0.9409 |
| Defect | 0.7581 | 0.8545 | 0.8034 |
| Accuracy |  |  | 0.9091 |
| Macro Average | 0.8581 | 0.8894 | 0.8721 |
| Weighted Average | 0.9146 | 0.9091 | 0.9110 |

Figure 8 shows the results of the confusion matrix for the proposed model. Figure 9 shows the evaluation of the model using the ROC curve. The closer the ROC curve area is to a value of 1, the better the model's performance. As is evident from the ROC curve, we achieved an area of 0.863 in this experiment.

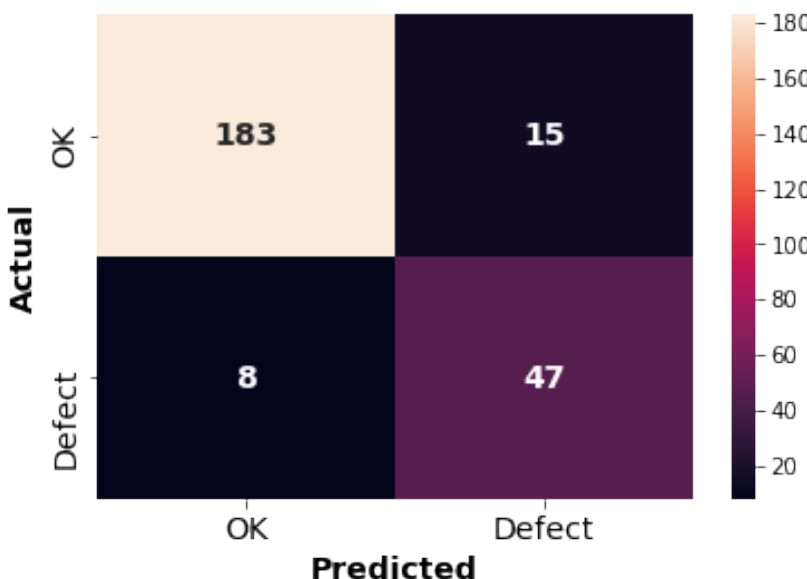

**Figure 8.** Confusion Matrix of Proposed Model.

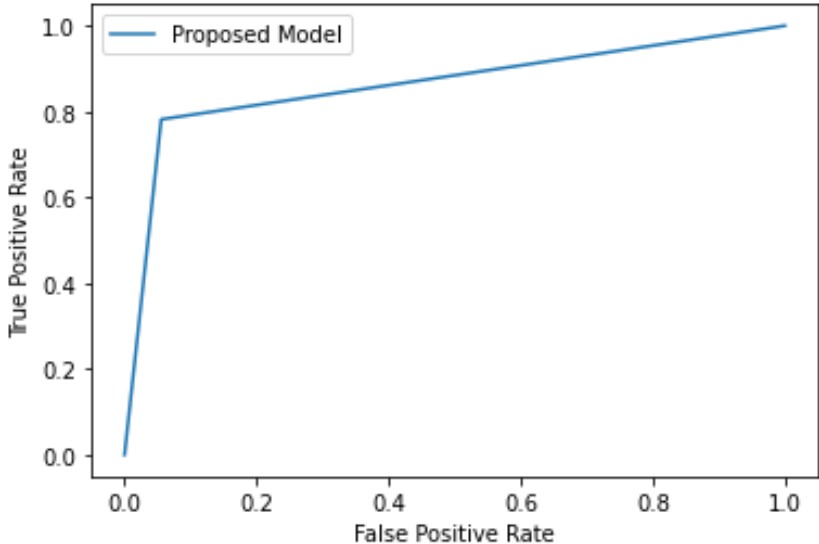

**Figure 9.** ROC Curve of Proposed Model.

Next, we experimented to optimize the built-in basic model. The first experiment was to increase the training data, the second was to alleviate the data imbalance in the test data, and the final one was to set the optimal threshold. First, we doubled the training data. Currently, we have increased the performance of the model by adding data obtained from the factory we are testing. Then, by adding abnormal data from the test data, the problem of imbalance between normal and abnormal data was solved to some extent. This is summarized in Table 6.

**Table 6.** Dataset of Case Model.

|  | Normal | Defect |
| --- | --- | --- |
| Training Data | 3428 | 400 |
| Validation Data | 632 | 200 |
| Test Data | 198 | 100 |

Figure 10 shows the results of the learning history of accuracy and loss in the training process. It can be seen that the training proceeds stably. This is thought to be due to the increase in training data. In addition, the model does not become an early stopping, and the validation loss continues to decrease as training progresses, so the training proceeds until the epoch reaches 50.

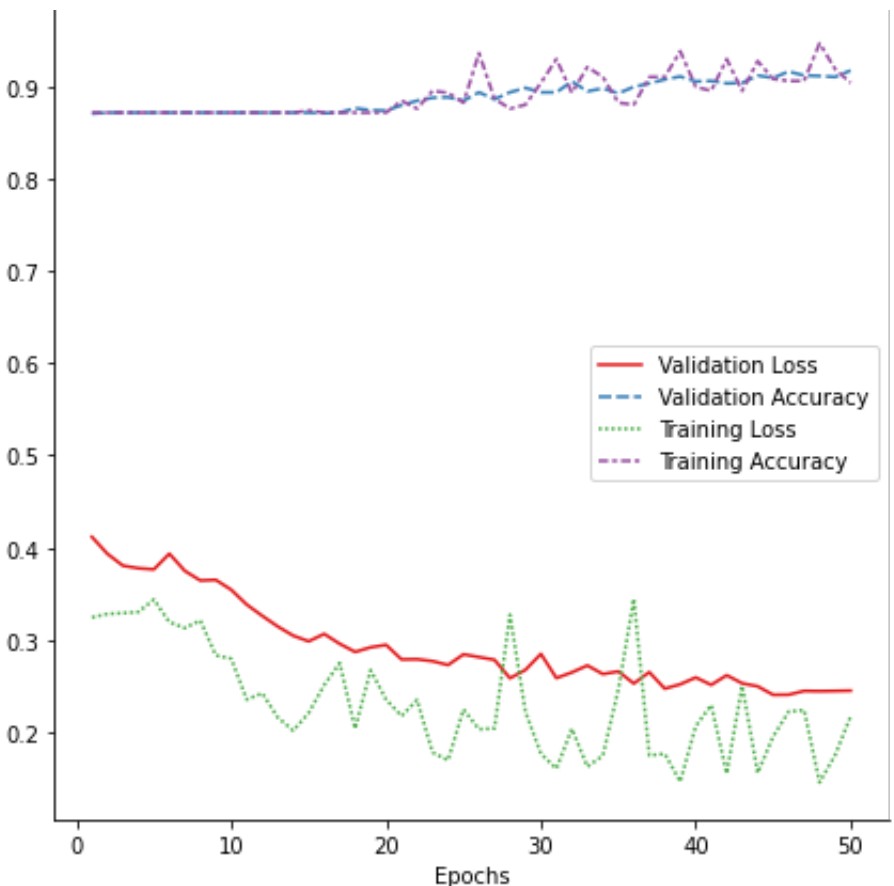

**Figure 10.** Learning History of Case Model.

Table 7 summarizes the case model's Precision, Recall, and F1-score values. We initially set the threshold to 0.5 in the softmax classification. However, when the threshold was set like this, defect data were judged to be normal in many cases. We found that the optimal threshold was 0.35 through repeated experiments. Figure 11 shows the results of the confusion matrix for the case model. Compared to the existing model, the F1 score increased from 0.9091 to 0.9262, and the prediction of actual defects is much better. The MCC score increased from 0.7311 to 0.8391, and the ROC AUC increased from 0.853 to 0.927, as shown in Figure 12.

**Table 7.** Results of Case Model.

|  | Precision | Recall | F1-Score |
|---|---|---|---|
| Normal | 0.9632 | 0.9242 | 0.9433 |
| Defect | 0.8611 | 0.9300 | 0.8942 |
| Accuracy |  |  | 0.9262 |
| Macro Average | 0.9121 | 0.9271 | 0.9188 |
| Weighted Average | 0.9289 | 0.9262 | 0.9268 |

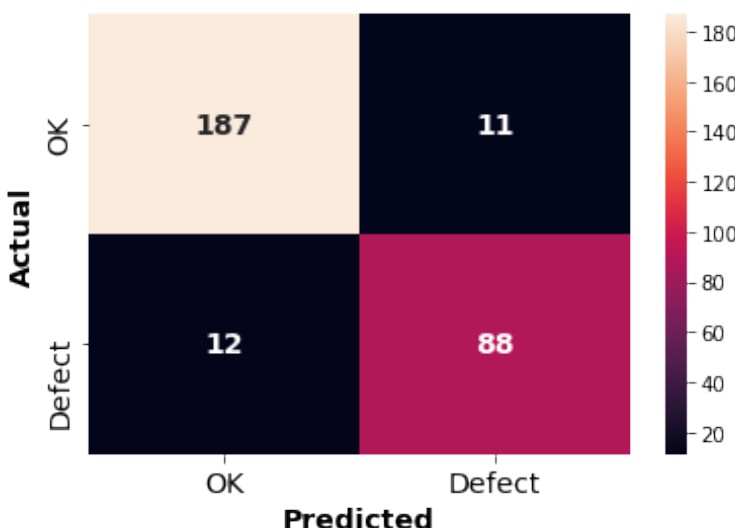

**Figure 11.** Confusion Matrix of Case Model.

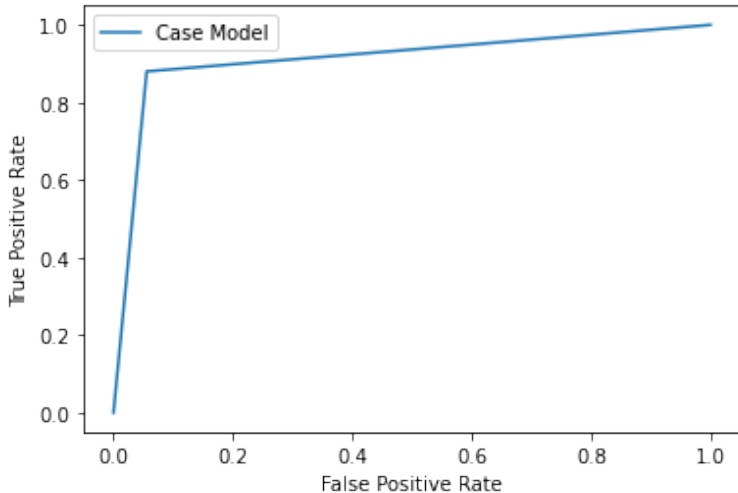

**Figure 12.** ROC Curve of Case Model.

## 5. Conclusions

To gain manufacturing competitiveness, the introduction of smart factories by SMEs is essential, but there are many difficulties in practical application. The early detection of injection molding defects plays an important role in identifying failures in the equipment. The development of an automated quality inspection model is drawing attention as a major component of the smart factory. However, injection molding has not received much attention in this area of research, because of product diversity, difficulty in obtaining uniform quality product images, and short cycle times. In this paper, we proposed a defect inspection system for injection molding in edge intelligence. By means of data augmentation, we solved the data shortage and imbalance problem of SMEs, introduced an actual smart factory method for the injection process, and measured the performance of the developed artificial intelligence model. In this study, we used a real case of introducing smart factories to SMEs in South Korea. We believe that this case can be further applied to similar injection molding processes. Furthermore, we measured the performance of the developed artificial intelligence model. The experiment showed that the accuracy of the proposed model was more than 90%, proving that the system can be applied in the field. In addition, we propesed methods to improve the accuracy of the model by conducting additional experiments.

In future work, we will study a method to detect defects based on bearing data provided by machinery equipped with an actual injection molding process. We will also study how to classify and, in consideration of mechanical factors, detect defects by further subdividing each type of defect. In addition, for the proposed method, because the noise cannot be completely removed, we will work on a better noise removal-method.

**Author Contributions:** Conceptualization, H.H. and J.J.; methodology, H.H.; software, H.H.; validation, H.H. and J.J.; formal analysis, H.H.; investigation, H.H.; resources, J.J.; data curation, H.H.; writing—original draft preparation, H.H.; writing—review and editing, J.J.; visualization, H.H.; supervision, J.J.; project administration, J.J.; funding acquisition, J.J. All authors have read and agreed to the published version of the manuscript.

**Funding:** This research was supported by the MSIT (Ministry of Science and ICT), Korea, under the ITRC (Information Technology Research Center) support program (IITP-2021-2018-0-01417) supervised by the IITP (Institute for Information & Communications Technology Planning & Evaluation) and the National Research Foundation of Korea(NRF) grant funded by the Korea government(MSIT) (No. 2021R1F1A1060054).

**Institutional Review Board Statement:** Not applicable.

**Informed Consent Statement:** Not applicable.

**Data Availability Statement:** Data sharing not applicable.

**Acknowledgments:** This research was supported by the MSIT (Ministry of Science and ICT), Korea, under the ICT Creative Consilence Program (IITP-2021-2020-0-01821) supervised by the IITP (Institute for Information & communications Technology Planning & Evaluation).

**Conflicts of Interest:** The authors declare no conflict of interest.

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
