# Peer review of "CNN-Based Defect Inspection for Injection Molding Using Edge Computing and Industrial IoT Systems"

_applsci, doi:10.3390/app11146378_

Round 1

Reviewer 1 Report

The paper is very easy to read. The configuration of the CNN is very well explained and the results are clearly positive. The weak part of this paper is the lack of interest to the readers. The number of images used in the training,2030 and 198 for testing data is not enough to conclude much either. I understand the difficulty of getting more images for training in this kind of system, but at the same time, this does not contribute with much even when the accuracy in training and testing is fairly high. Overall,  I'm not sure if a system like this can be used to detect defects in injection molding processes. 

Reviewer 2 Report

Overall, this paper is interesting, well structured and self-contained. I think that the contribution herein introduced may have several and useful practical implications. The paper appears to be sound and relies on well-know building blocks. Again, it is written by using an appropriate technical language.

I want suggest this paper to authors:

Lombardi, M.; Pascale, F.; Santaniello, D. Internet of Things: A General Overview between Architectures, Protocols and Applications. Information 2021, 12, 87. https://doi.org/10.3390/info12020087

Finally, in some parts, the clarity and editorial quality of the paper weaken. As a consequence, such parts result to be quite difficult to read. Therefore, I would suggest to carefully improve the prose of writing in order to make this paper easier to read.

Reviewer 3 Report

The paper addresses an important question of fault product detection on a production line (mould injection process) by introducing a novel approach, combining IoT, edge computing and CNN. Authors aimed to present this novel approach on a SME case.

The manuscript is very difficult to read because the language is very unclear and the typical IMRAD structure has not been followed. There are many mistakes in referencing, in many places references are missing (and should be there), only 20% of the references are from the past three years, which is a bit odd since the topic has been researched most intensively in the past years.

The major flaw of the paper is that it lacks the methodology section completely. It is unclear what the result of the study is: is it the development of an artefact (a novel methodology) - so the design science research approach or is it an experiment. The experiment is mentioned in one sentence, but the experiment design, the variables, hypotheses are not. To sum up, there is no methodological approach described, therefore it is difficult to assess what has been done and whether the goals of the study had been achieved.

Further, the results section has two subsections, 4.3 Results and 4.4 Results - why is that? Both subsections start with the figure (eg. Figure 6) and the subsection 4.4 starts with the figure and a table. It is not advisable to start sections with figures and/or tables, it is more difficult to read.

Results are rather scarce compared to the introduction and literature review part of the text -  only 2 pages with very little text – 4 figures and one table and there is no discussion of the results. The scarce results are followed by a short conclusions, which are very general and do not provide the conclusion to whether the problem was solved, what the novelty is compared to the previous literature. Finally, authors line up their future work without arguing why this steps need to be taken.

Some parts of texts are really difficult to understand, for example:

Line 16, 17: there is a punctuation where one would think is one sentence.  »Defect inspection is very important process in these injection molding processes. because it can reduce the risk and cost of providing defective products to customers.«

Lines 23-25: »Repeating this process is so high that the risk of financial losses throughout the industry that the urgency of automating surface defect detection is expanding to manufacturing. Causes worker fatigue due to repetitive work.« - Unclear.

In several places, the authors state »there are many studies ...« and don't reference them. One would expect that all the mentioned studies are referred to. For example:

Line 25-28: »To address these issues, many studies have been conducted on automation and defect detection. Among them, research on building a smart factory using the Internet of Things (IoT) is actively underway.«

Lines 33-36: Two practically identical sentences. »Edge computing is defined as long-term cloud computing (CC) where data is calculated near the edge of the network where data is generated. Edge computing is defined as long-term cloud computing (CC) in which data is calculated closer to the edge of the network in the place of data production.«

Line 43-44: acronyms are first introduced in full text with the acronym in bracket and can further be used as an acronym. There are a total confusion in using acronyms in the manuscript. Further the reference [4] probably belongs to the previous section? This too is inconsistent throughout the text. Further, no reference to the ImageNet Challenge – unless it is the [4], which is not clear.  »[4] In particular, the development of the CNN (Convolutional Neural Networks) model achieved higher accuracy than humans perceived in the ImageNet Challenge.«

Line 50: »data balance is the problem« - it would be beneficial if the authors described the data imbalance problem of this particular case in greater detail.

Lines 51-52: »Finally, Currently, defect detection is being carried out in a plane.«  - The sentence is unclear.

Lines 58-59: »Various data augmentation methods have been studied and applied to solve the problem of lack of data in the field. A typical example is the medical field.« - With no reference, very imprecise writing.

Lines 118-119: »The filter moves horizontally and vertically in the input image, performing convolution and

119 activation function operations, extracting features, and displaying Feature Map. Yield.« What is meant by this sentence or two sentences?

Line 122: » by Recun et al., ...« I believe the referenced name is incorrect, however no reference is added in brackets.

Line 237: »Below examples are ...« - Which examples?

Lines 251-252: »After that, the architecture would completed with a softmax layer at the end for defect determination.« - sentence?

Line 254: Experiment and Results – where is the experiment described?

Lines 259-260: »The company is producing female products, which called tampons.« - First, the sentence is incorrect grammatically, second, is the production of tampons by injection moulding?

Lines 263-264: »And 20 % of the training data was used as verification data during training.« - do you mean the verification data set – validation or test data set? When calculated from Table 3, the test set does not equal to 20%.

In section 4.2 authors present the evaluation metrics in detail, but this brings no added value to the paper, since this measures are generally known.

Lines 297-299: »Epoch was set to 50, and training was stopped if validation loss did not improve after 10 or more epochs were repeated. As a result, 16 epoch results were obtained.« - The sentences are unclear.

Lines 305-306: »The ability to predict the normal product from the normal person data is good, ...« - What did you mean here?

Lines 325-327: »In this study, by introducing the actual case of introducing smart factory to SMEs in Korea, we introduce advanced cases that will be additionally applied to similar injection modling cases.« - Where did you introduce these cases?

These are just some of the examples of non-clarity, incorrect referencing and other things that really distract the reader from the content. It is thus difficult to evaluate whether the manuscript has potential or not. In the present state, I don't recommend this manuscript to be accepted for publication.

Reviewer 4 Report

The research work presents the design and application of Convolutional Neural Networks to the Injection Molding Process which relies on the Edge Computing paradigm.

The topic is interesting and the paper is well structured but the quality of writing is really poor. It undermines the quality of the entire paper. An extensive editing is required before resubmitting the paper. I suggest the authors to look for someone with high proficiency in english and request help for revising the paper.

I found some statement that are not clear:

  • row 159 "This is why cloud computing is not always efficient in processing data when it is generated at the edge network" ...as far as I know in an IoT systems most of the data are generated at the edge and transferred to the cloud so the cloud is almost in every case processing data coming from the edge of the network.
  • row 223 and 224. "Performing deep learning methods that do not have sufficient training data to train parameters usually results in overfitting problems". As far as I know, having not enough data to be used for training causes underfitting problems and not overfitting. 

References are badly reported in the text such as row 123 and 127. The first reference comes immediately at the beginning of a sentence and the second one reports citations between dots ". [29-30] ." I wonder what is the meaning of that.

No mention is done about the software and tools used for conducting the experiment. I wonder which tool have been used. I also wonder if data could be shared since their absence affects replicability of the experiment.

The work reads as a nice Deep Learning exercise but it lacks details for actually replicating the experiments.

The results are poorly described; they are just reported without really discussing their implication for the specific case study.

I suggest the authors to carefully revise the paper, especially for what concerns the usage of English. 

Round 2

Reviewer 3 Report

Dear authors,

although you have improved the structure and text, there are still many mistakes found in the text, including in parts when your answer to reviewer was that you have corrected it (e.g. the reference [25]). The answer that tampons are produced by injection molding - Tampons are, to my knowledge made of cotton, and are not injection molded. Maybe I am wrong, but I would guess that the tampon package/aplicquator might be the one product that is injection molded? You have rewritten the sentence, but the new one is still not clear. "In this study, using the actual case of introducing smart factories to SMEs in Korea, we introduced advanced cases that will be additionally applied to similar injection modeling cases."

It is very difficult to assess the paper where there are lots of mistakes, especially when it is the second round of review. I would strongly suggest to polish the manuscript before sending it back to the review process. 

Reviewer 4 Report

The research work has been slightly improved by addressing some comments.

However,  there are still some shadows concerning the conducted experiments.

No software tool is mentioned in the entire paper. I really would like to know which software tools have been used to conduct the experiment.

Ok that the data are confidential but an example (even without real data) could help a reader to better understand which kind of data the approach can process.

The overall "story" described in the paper is ok but the experiments are not convincing
